# Effects of Impregnated Amidophosphonate Ligand Concentration on the Uranium Extraction Behavior of Mesoporous Silica

**DOI:** 10.3390/molecules27144342

**Published:** 2022-07-06

**Authors:** Aline Dressler, Antoine Leydier, Agnès Grandjean

**Affiliations:** CEA, DES, ISEC, DMRC, University Montpellier, 34000 Marcoule, France; aline.dressler@cea.fr (A.D.); agnes.grandjean@cea.fr (A.G.)

**Keywords:** functionalized silica, hybrid material, uranium, effluent, extraction, capacity, sulfate

## Abstract

A series of solid-phase uranium extractants were prepared by post-synthesis impregnation of a mesoporous silica support previously functionalized with octyl chains by direct silanization. Five materials were synthesized with 0, 0.2, 0.3, 0.4 and 0.5 mmol of the amidophosphonate ligand DEHCEBP per gram of functionalized solid, and the effect of the ligand concentration on the uranium extraction efficiency and selectivity of the materials was investigated. Nitrogen adsorption–desorption data show that with increasing ligand loadings, the specific surface area and average pore volume decrease as the amidophosphonate ligand fills first the micropores and then the mesopores of the support. Acidic uranium solutions with a high sulfate content were used to replicate the conditions in ore treatment leaching solutions. Considering the extraction kinetics, the equilibration time was found to increase with the ligand concentration, which can be explained by the clogging of micropores and the multilayer arrangement of the DEHCEBP molecules in the materials with their highest ligand contents. The fact that the equilibrium ligand/uranium ratio is about 2 mol/mol regardless of the ligand concentration in the material suggests that all the ligand molecules remain accessible for extraction. The maximum uranium extraction capacities ranged from 30 mg∙g^−1^ at 0.2 mmol∙g^−1^ DEHCEBP to 54 mg∙g^−1^ in the material with 0.5 mmol∙g^−1^ DEHCEBP. These materials could therefore potentially be used as solid-phase uranium extractants in acidic solutions with high sulfate concentrations.

## 1. Introduction

Uranium recovery and purification processes differ depending on the mineralogy and lithology of the host rock and processing options but generally include comminution and leaching, followed by purification and precipitation for further refining. The leaching solvent is usually an acid solution, most often sulfuric acid because of its relatively low cost and wide availability. Purification then usually involves solid-phase and/or solvent extraction, to selectively concentrate the uranium present in the leaching solution [1,2,3].

Solid-phase extraction (SPE) is an attractive alternative to traditional liquid–liquid extraction (LLE) for the selective recovery of low-concentration uranium from acidic aqueous phases such as uranium ore leaching solutions, notably because of lower costs, processing times and solvent use [4,5]. SPE is also a more compact process and offers the possibility of performing extraction and back-extraction in separate locations. For example, ores can be extracted on mining sites and then back-extracted into a purification center after transportation. Processes based on solid-phase extractants such as ion exchange and the resin-in-pulp process are already used in the uranium industry, but improved SPE supports are required to manage more highly concentrated leaching solutions and to increase selectivity [3]. A well-investigated SPE strategy for the removal of uranium ions from aqueous effluents has been to functionalize metal oxide particles [6], mesoporous silica [7,8], carbon supports [9,10,11], MOFs [12,13], fibers [14,15] and resins [3,16,17]. 

Mesoporous hybrid silica is particularly interesting in this context because of the ease with which its morphology and surface properties can be modified, and silica-based materials have already been investigated for the adsorption of organic and inorganic pollutants [18,19]. 

We have previously studied the post functionalization of silica with carbamoylalkylphosphonates [20,21,22,23,24], bifunctional amidophosphonate ligands that have also been used for the solvent extraction of uranium from (acidic) phosphate [25,26] and sulfate solutions [27]. These ligands were attached to the silica surface by chemical bonding (grafting) or by simple physical adsorption (impregnation). The uranium extraction efficiency of the grafted material was improved by optimizing the length and the steric hindrance of the alkyl chain of the carbamoylalkylphosphonate moieties [23] and the initial pore size of the support [20].

In the most recent of these studies, we showed that materials functionalized by impregnation have a higher extraction efficiency and selectivity at the high sulfate concentrations typical of uranium ore treatment solutions [24]. EXAFS data suggested that to be extracted by the phosphonate ligand, uranyl species have to be desulfurized, a process that appears to be more energetically favorable for the impregnated ligands than for the grafted ones. 

The effects of impregnation and grafting have previously been compared in monodisperse porous polymer particles [28], mesoporous silica [29] and hollow silica microspheres [30]. Song et al. [28] compared impregnated and grafted materials with the same organic loads to extract strontium from acidic solutions. The impregnated material was found to have a higher extraction capacity and this was attributed to the absence of conformational constraints in forming complexes with strontium. The adsorption kinetics were slower than with the grafted material, however [28]. Elsewhere, silica-based materials have been functionalized with amines for CO_2_ adsorption [29,30]. The impregnated adsorbents were found to have higher CO_2_ adsorption capacities and amine loading efficiencies. In the same context, Monte Carlo simulations have shown that impregnation leads to higher adsorption capacities than grafting because the functionalized chains are more mobile where they are impregnated rather than grafted [31]. 

Selective high-capacity adsorbents are highly sought after for a range of practical applications. An effective means of improving the uranium extraction performance of hybrid materials is to increase the surface concentration of ligand molecules. However, while in liquid–liquid processes the ligands are fully accessible, in SPE the accessibility of the ligands strongly depends on their arrangement inside the pores [32] and the mesostructure of the solid phase (here, silica). Excess organic groups can become sterically hindered within the pores and form agglomerates between particles, both of which limit the diffusion of the target species within the impregnated porous matrix [32,33].

The extraction of uranium from sulfate solutions at low pH (<3.0) has already been reported in previous studies and generally shows extraction capacities between 20 and 35 mg/g [20,23,24,34]. In order to obtain materials with higher uranium extraction capacity, the present study investigates how the loading of a carbamoylalkylphosphonate ligand into mesoporous silica affects the uranium extraction efficiency. The ligand was impregnated at various concentrations onto silica supports pre-functionalized with alkyl chains and its effects were interpreted in terms of the arrangements of the molecules on the surface of the pores. The selectivity of the synthesized materials was studied in high sulfate solutions containing a few hundred mg/L of uranium—conditions representative of ore treatment solutions—and Fe and Mo as competing ions, as these ions are known to be the most problematic for amidophosphonate ligands [27].

## 2. Results and Discussion

### 2.1. Structural Characterization

The correlation between the uranium extraction efficiency of the materials and their various ligand concentrations was investigated through different parameters: the chemical structure and concentration of the amidophosphonate ligand, and the change in specific surface area and pore volume of the materials after functionalization. 

As reported previously [23,24], the ^29^Si CP-MAS NMR spectra of the pre-functionalized materials showed that the octyl moieties are mainly linked to the silica surface via T2 units (C_8_-Si(OSi)_2_(OCH_2_CH_3_) bonds, implying the presence of ethoxy groups) and T3 units (C_8_-Si(OSi)_3_) [35]. The grafting of the alkyl group was also confirmed by ^13^C solid-state NMR.

Along with the NMR data, the peak at 1635 cm^−1^ (C=O stretching, amide) in the IR spectra of the impregnated materials (Figure 1) confirms the presence of carbamoylphosphonate molecules on the surface of the silica particles. All the ^31^P CP-MAS NMR spectra consisted of a single peak at around 26 ppm, as observed previously for materials prepared in this way [24], indicating that the chemical structure of the ligand does not change within the studied concentration range (0.2–0.5 mmol∙g^−1^).

Figure 2 shows the nitrogen adsorption–desorption isotherms of the pre-functionalized silica support and of the functionalized materials with different ligand loadings. The isotherms are all type IV, with an H2 hysteresis loop [36]. Since C_8_@D60 has a total pore volume of 0.68 cm^3^∙g^−1^ and the density of DEHCEBP is about 1.0 g∙cm^−3^, the maximum ligand concentration in this support is 40 wt%. However, samples prepared with more than 0.5 mmol∙g^−1^ DEHCEBP were found to be doughy and sticky, indicating that some of the ligands were located outside the pores and that the maximum loading using this procedure is about 0.5 mmol∙g^−1^ or 20 wt%.

The ligand concentrations, specific surface areas and pore volumes of the studied materials are listed in Table 1. The organic content was calculated from the weight loss between 150 and 950 °C obtained from TGA. This temperature range was chosen to avoid including the evaporation of physisorbed water, the ligand itself remaining thermally stable up to 150 °C (Appendix A). The measured ligand concentrations are in agreement with the values expected from the initial amounts used during the impregnation step, confirming the robustness of the synthesis route. The BET-specific surface area and total pore volume decrease after functionalization and are inversely related to the ligand concentration, suggesting that at least some of the ligand molecules are adsorbed within the pores.

The pore size distributions of the impregnated materials (Figure 3) show that the total pore volume is inversely related to the ligand concentration and that the smaller diameter pores are almost completely filled in Imp-0.5/C_8_@D60. The mean pore diameter decreases from Imp-0.2/C_8_@D60 to Imp-0.4/C_8_@D60 but is similar in Imp-0.4/C_8_@D60 and Imp-0.5/C_8_@D60. This may be because at high ligand concentrations, the ligand molecules become blocked in the partially filled micropores during impregnation, and cannot diffuse to the mesopores, leading to the preferential filling of the former. Figure 4 shows indeed that the percentage of filled micropore volume increases from 58 to 75 vol% between Imp-0.4/C8@D60 and Imp-0.5/C_8_@D60, whereas the filled mesopore volume only increases from 52 to 58 vol%. The fact that the percentage of filled micropore volume is in all cases higher than the percentage of filled mesopore volume also suggests that the DEHCEBP molecules first fill the micropores and then the mesopores in the support.

Table 1 also shows that the volume occupied by a DEHCEBP molecule is practically the same in all the synthesized materials, indicating that no denser layers are formed at higher ligand concentrations. This is consistent with the fact that the total pore volume decreases linearly with the ligand concentration (Figure 5). 

Assuming that one DEHCEBP molecule is about 2 nm long (based on the Tanford formula, Equation (4)) and that the movement of the impregnated ligands is not restricted, each molecule occupies a circular area of about 3.3 nm^2^. The theoretical surface density required to saturate the surface of the solid support with one layer of ligand molecules is then about 0.3 nm^−2^, suggesting that the DEHCEBP molecules are arranged on the surface of the support in a single layer in Imp-0.2/C_8_@D60 and Imp-0.3/C_8_@D60 materials, and in multiple layers in Imp-0.4/C_8_@D60 and Imp-0.5/C_8_@D60.

In summary, these results indicate that at concentrations up to 0.3 mmol∙g^−1^, DEHCEBP molecules form monolayers on the pre-functionalized surface of the support, filling first the micropores and then then the mesopores of the structure. At higher concentrations, up to 0.5 mmol∙g^−1^, the ligands adsorb in multiple layers and nearly completely fill the micropores of the silica support.

### 2.2. Extraction of Uranium from Sulfuric Acid Solutions

The uranium extraction capacity (Q_U_, mg∙g^−1^) of the four synthesized materials was evaluated after 2 h and 1, 7, 14 and 21 days in a high sulfate concentration solution ([SO_4_]^2−^/[U] = 900 mol/mol) with and without iron and molybdenum as competing cations (Table 2). Since previous extraction experiments performed under similar conditions with pure and pre-functionalized silica [24] showed that the silanol, methoxy and octyl groups in these materials are not involved in the extraction process, the uranium extraction capacity of the functionalized materials is due entirely to the presence of amidophosphonate ligands.

Figure 6 shows that the equilibration time increases with the ligand load in both solutions (with and without iron and molybdenum as competing cations). This is in keeping with the above results that at higher concentrations, the ligands form multiple layers on the surface of the materials and fill the micropores, therefore slowing the extraction kinetics. At equivalent ligand concentrations, the equilibration times are shorter in the presence of competing ions (Figure 6b versus Figure 6a), because higher salt concentrations facilitate the penetration of uranyl ions into organic layers, mass transfer being driven by the concentration gradient between the solution and the ligand-functionalized surface. This driving force increases with the salt concentration in the solution.

Figure 7a shows that at equilibrium in the absence of competing cations, about two DEHCEBP molecules are required to extract one uranyl (VI) ion, regardless of the concentration of ligand molecules on the surface of the adsorbents. This suggests that all the ligand molecules remain accessible for extraction and explains why the maximum uranium extraction capacity of the materials increases with the DEHCEBP concentration. The values measured after 21 days ranged from 30 mg∙g^−1^ in Imp-0.2/C_8_@D60 to 54 mg∙g^−1^ in Imp-0.5/C_8_@D60 (Table 3). Charlot et al. [21] measured the same ligand/uranium ratio for materials grafted with DEHCEBP in sulfuric media at pH 3 with a low sulfate/uranium ratio ([SO_4_]^2−^/[U] = 50 mol/mol). We can thus conclude that the extraction mechanism in these impregnated materials with up to 0.5 mmol∙g^−1^ DEHCEBP at high sulfate concentrations is the same as described by Charlot et al. [21] for the grafted materials. 

In the presence of iron and molybdenum (Figure 6b and Figure 7b), the uranium extraction capacities of the materials are substantially lower (about 75% of the values measured in the absence of competing cations, Table 3) and the equilibrium ligand/uranium ratio is higher (~2.6). The same trend was observed by Charlot et al. in solutions with a low sulfate concentration at pH 3 [21] and is probably due to the extraction of some of the iron and/or molybdenum in the solution instead of uranium.

Nevertheless, all the impregnated materials were highly selective for U versus Fe, with selectivity factors (Equation (7)) greater than 60. (Accurate values could not be determined because the variations in the iron concentration before and after adsorption were smaller than the associated measurement errors.) The Imp-0.2/C_8_@D60 and Imp-0.3/C_8_@D60 materials were similarly selective versus Mo, with variations within the ICP-AES error margin. Imp-0.4/C_8_@D60 and Imp-0.5/C_8_@D60 extracted measurable amounts of molybdenum (about 0.5 mg∙g^−1^), but these materials were nevertheless highly selective for U versus Mo with $SF_{Mo}^{U}$ = 9 and 11, respectively.

## 3. Materials and Methods

### 3.1. Chemicals

All organic reagents were used as received from Sigma-Aldrich (St. Louis, Missouri, USA), Acros Organics (Geel, Belgium) and Fluka (Buchs, Switzerland). Solvents were purchased from Acros Organics, Pro-Labo (Tokyo, Japan), Fluka, and Sigma-Aldrich. Anhydrous solvents were obtained from Acros Organics.

### 3.2. Materials Synthesis

The organic ligand Di-2-EthylHexylCarbamoyleEthylButyl Phosphonate (DEHCEBP) was synthesized as described by Turgis et al. [37]. Commercial mesoporous silica particles (Davisil^®^, Sigma-Aldrich; pore size, 60 Å) were then functionalized by wet impregnation in two steps (Figure 8). The silica supports were first pre-functionalized with octyl chains by direct silanization (for 30 min at 200 °C in toluene under 850 W microwave irradiation), using commercial triethoxyoctylsilane (Sigma-Aldrich), yielding C_8_@D60. The supports were then impregnated as follows: about 2 g of previously pre-functionalized silica (C_8_@D60) was mixed with 20 mL of dichloromethane containing the desired amount of the DEHCEBP in a 50 mL round bottom flask for 3 h. The final materials—Imp-*X*/C_8_@D60, where *X* (0.2, 0.3, 0.4. or 0.5 mmol/g) is the amount of DEHCEBP per unit mass of functionalized solid—were obtained after evaporation of the solvents and 12 h of vacuum drying.

The characteristics of the ligand have already been reported elsewhere [24].

### 3.3. Uranium-Containing Solutions

Solutions with a few hundred mg/L of uranium and a high sulfate concentration ([SO_4_]^2−^/[U] = 900 mol/mol) were prepared to simulate the conditions in ore leaching solutions (Table 2). The effects of competing ions on the uranium extraction capacities of the studied materials were analyzed using solutions containing Fe, added as iron sulfate (Fe_2_(SO_4_)_3_·3H_2_O), and Mo, added as sodium molybdate (Na_2_MoO_4_). The pH was adjusted using sulfuric acid and the desired [SO_4_]^2−^/[U] ratio was achieved by adding sodium sulfate.

### 3.4. Characterization of the Organic Ligand

Solution ^1^H, ^31^P and ^13^C NMR spectra were recorded on a Bruker 400 ultrashield VS spectrometer (Larmor frequencies, 400.13 MHz for ^1^H, 161.976 MHz for ^31^P, 100.613 MHz for ^13^C) using deuterated chloroform as the solvent and internal standard. 

### 3.5. Characterization of the Materials

Ligand concentrations per unit mass (τ_L_, mmol∙g^−1^) were determined by thermogravimetric analysis (TGA/DSC 1, Mettler Toledo), as described and validated by Charlot et al. [20] by comparison with carbon and nitrogen elemental analysis. About 20 mg of material was placed in a 70 µL alumina pan and heated from 303 to 1273 K at 5 K∙min^−1^ under 30 mL∙min^−1^ air flow. The ligand concentrations were calculated using Equation (1):(1)$$\tau_{L}=\frac{\Delta W_{f}\left(\%\right)-\Delta W_{i}\left(\%\right)}{\;M_{L}*\;\Delta W_{L}\left(\%\right)}$$
where Δ*W_f_(%)* and Δ*W_i_(%)* are, respectively, the percentage weight losses between 150 and 950 °C of the final functionalized support and of the pre-functionalized silica material (C_8_@D60), and *M_L_* is the molar mass of the organic ligand. The weight loss of the pure amidophosphonate ligand (Δ*W_L_(%)*) was also determined under the same conditions to obtain a correction factor, assuming that some non-volatile phosphorus species must form during the calcination of DEHCEBP. This correction factor was added to avoid any underestimation of the ligand content. 

The presence of the ligand on the surface of the silica particles was verified by attenuated total reflection (ATR)-FTIR analysis and solid-state cross-polarization magic-angle spinning (CP-MAS) NMR spectroscopy. The ATR-FTIR experiments were performed with a Nicolet iS50 device equipped with a diamond crystal plate ATR element. A background spectrum was recorded before each experiment so that the contributions of carbon dioxide, water vapor and diamond crystals could be subtracted from the spectrum. Each spectrum was obtained as an average of 32 scans in the range 400–4000 cm^−1^ at 4 cm^−1^ resolution. ATR corrections were applied during processing. ^13^C, ^31^P and ^29^Si CP-MAS NMR spectra were recorded on a 400 MHz Bruker Avance spectrometer. The samples were spun at 10 kHz in a 4 mm zirconia rotor. 

Nitrogen adsorption–desorption isotherms were measured at −196 °C using a Micromeritics ASAP 2020 surface area and pore size analyzer. The samples were degassed at 90 °C for 24 h before analysis. Specific surface areas were calculated using the Brunauer–Emmett–Teller (BET) method. Pore size distributions were calculated using the Barrett–Joyner–Halenda (BJH) method. Total pore volumes were determined using the volume of adsorbed gas at P/P_0_≈1. Micropore volumes were measured at P/P_0_ = 0.05 and mesopore volumes were then deduced from (total pore volume)—(micropore volume), to estimate the percentage of the volume of both pore types occupied by the ligand.

The volume of one DEHCEBP molecule was calculated from the measured total pore volumes of the final materials and their respective ligand loads, using
(2)$$V_{L}=\frac{\Delta V_{pores}}{\tau_{L}*N_{A}}*10^{21}$$
where Δ*V_pores_* (cm^3^∙g^−1^) is the difference between the pore volume of the pre-functionalized support and the respective adsorbent after impregnation, *τ_L_* is the DEHCEBP concentration (mol∙g^−1^) (Equation (1)) and *N*_A_ is the Avogadro number.

The surface density of the ligand (DEHCEBP∙nm^−2^) was calculated from the BET-specific surface area S_BET_ (m^2^∙g^−1^) of the pre-functionalized support and the organic content of the final materials using Equation (3):(3)$$d_{SLigand}=\frac{\tau_{L}*\;N_{A}}{S_{BET}*10^{18}}$$

The area occupied by a single DEHCEBP molecule was estimated using the empirical Tanford formula for the length (*r*, in nm) of a simple hydrocarbon chain of *n* atoms:(4)r=0.154+0.1265n

### 3.6. Uranium Extraction Experiments

The equilibration time was estimated from extraction tests conducted in batch mode at 25 °C with shaking for between 2 h and 21 days. The experiments were performed in the absence and presence of competing ions to separately determine their impact on the equilibration time, maximum uranium extraction capacity and selectivity. All experiments were performed with solid/liquid ratios (Ψ_S/L_) of about 5 mg∙mL^−1^. Each point in the kinetics curves corresponds to a single experiment.

After the chosen contact times, the materials were separated from the liquid phase by filtration through a 0.22 µm cellulose acetate membrane. The concentrations of ions in the liquid phase before and after contact with the material were measured by inductively coupled plasma atomic emission spectroscopy (ICP-AES; 2% nitric acid; Analytik Jena PlasmaQuant PQ 9000). These values were then used to calculate the extraction capacity and selectivity of the materials. The extraction capacity at equilibrium (*Q_X,e_*, the mass or amount of uranium extracted per unit mass of solid) was calculated using Equation (5):(5)$$Q_{X}=\left[X_{i}-X_{eq}\right]*\;\frac{V}{m\;}$$
where *X_i_* and *X_eq_* (mg∙L^−1^ or mmol∙L^−1^) are, respectively, the uranium concentrations in the solution before contact and at equilibrium between the solid and the solution, *V* is the volume of the solution (L) and m is the mass of the solid sample (g).

The measured concentrations of ligand (τ*_L_*, mmol∙g^−1^) inside each solid support were used to determine the ligand to uranium molar ratio (*L/U*) for each material under these experimental conditions (Equation (6)):(6)$$L/U=\frac{\tau_{L}}{Q_{U}}$$

If all ligand molecules are accessible for extraction, *L/U* corresponds to the stoichiometric coefficient of the complexes formed inside the pores of the material during extraction.

The selectivity factor (SF) of the materials for uranium in the presence of Mo and Fe as competing ions was calculated using Equation (7):(7)$$SF_{Fe\;or\;Mo}^{U}=\frac{\frac{Q_{U}}{Q_{Fe\;or\;Mo}}}{\frac{{\left[U\right]}_{i}}{{\left[Fe\right]}_{i}\;or\;{\left[Mo\;\right]}_{i}}}$$
where [*U*]*_i_*, [*Fe*]*_i_* and [*Mo*]*_i_* are the initial concentrations of the respective cations in the solution. Selectivity factors greater than 1 indicate that the material is selective for uranium, whereas values less than 1 mean the material has a higher affinity for the competing ion.

## 4. Conclusions

A series of adsorbents were prepared by post-synthesis impregnation of a mesoporous silica support with different loadings of an amidophosphonate ligand (DEHCEBP). The support was pre-functionalized with octyl chains by direct silanization. Nitrogen adsorption–desorption measurements revealed that this impregnation procedure fundamentally alters the porous structure of the support as the organic ligand fills the micropores and then the mesopores in the support. The mean pore size is inversely related to the ligand concentration between 0.2 and 0.4 mmol∙g^−1^ but does not decrease further between 0.4 and 0.5 mmol∙g^−1^, presumably because free ligand molecules tend to become blocked in the ligand-filled micropores and are therefore less likely to be adsorbed in the mesopores. The total pore volume decreases linearly with the ligand loading, most probably because of the formation of multilayers at high concentrations of DEHCEBP, since no densification of the ligand at increasing concentrations was observed.

In uranium extraction tests performed in simulated ore leaching solutions, the equilibration time increased with the ligand loading of the material, in keeping with a clogging of the micropores and a multilayer arrangement of the ligand molecules in the materials with higher ligand contents. The extraction kinetics were faster in the presence of competing cations (Fe and Mo), with the higher salt concentration facilitating the penetration of uranyl ions into the organic layers. The synthesized materials were all highly selective for U versus Fe and Mo, but the uranium extraction capacity was about 25% lower, due to the competitive interaction of these ions with the ligand molecules. The ligand to uranium molar ratio reached the same value at equilibrium regardless of the concentration of DEHCEBP molecules on the surface of the adsorbents, as reflected by the increase in the maximum uranium extraction capacity up to 54 mg∙g^−1^ in the material with highest ligand concentration. These materials could therefore be used as solid-phase uranium extractions in acidic solutions with high sulfate concentrations and their amenability to recycling, elution and reuse is currently being studied.

## Figures and Tables

**Figure 1 molecules-27-04342-f001:**
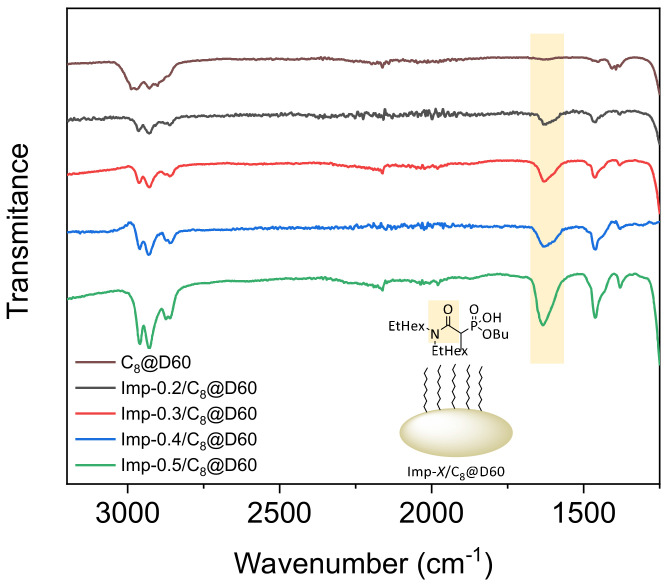
Fourier-transform infrared spectra of the pre-functionalized support (C_8_@D60) and the impregnated materials with different DEHCEBP loadings (0.2−0.5 mmol∙g^−1^).

**Figure 2 molecules-27-04342-f002:**
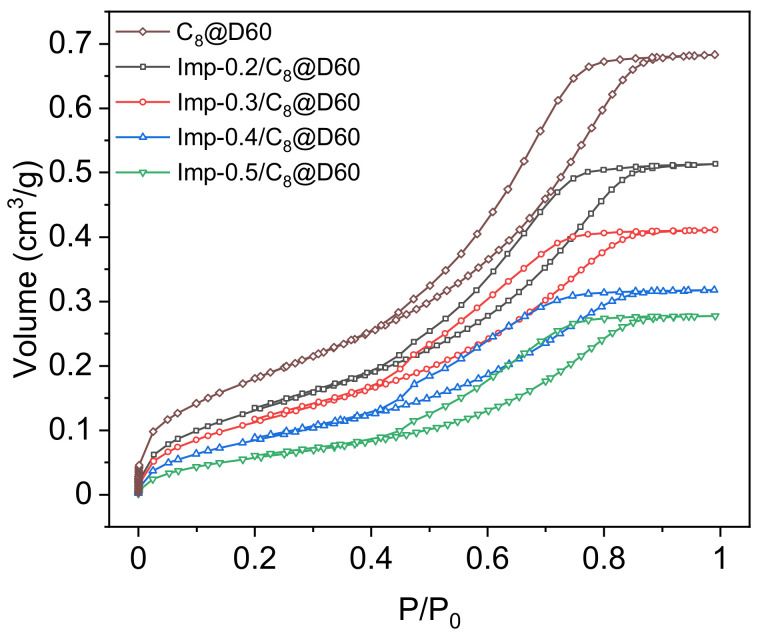
Nitrogen adsorption–desorption isotherms of the pre-functionalized support (C_8_@D60) and the impregnated materials with different DEHCEBP loadings (0.2–0.5 mmol∙g^−1^).

**Figure 3 molecules-27-04342-f003:**
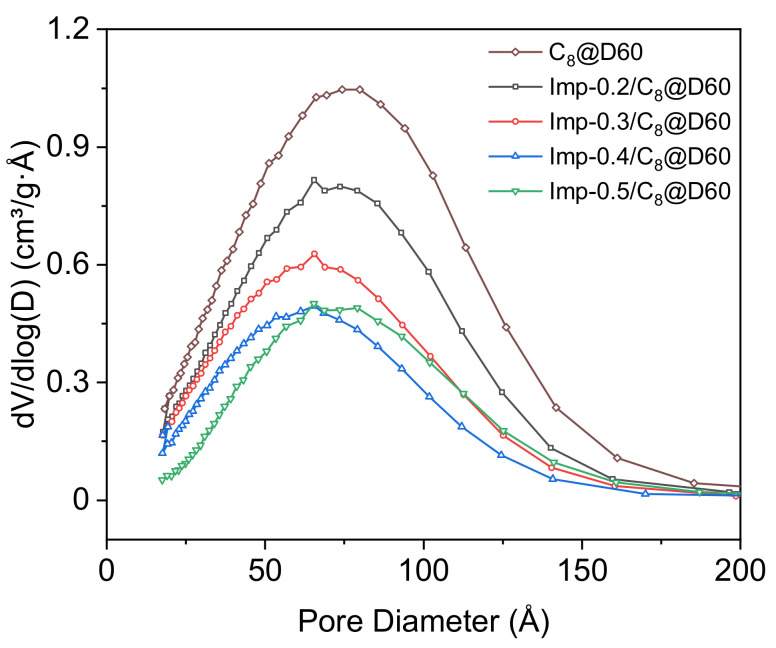
Pore size distributions of the pre-functionalized support (C_8_@D60) and the impregnated materials with different DEHCEBP loadings (0.2–0.5 mmol∙g^−1^).

**Figure 4 molecules-27-04342-f004:**
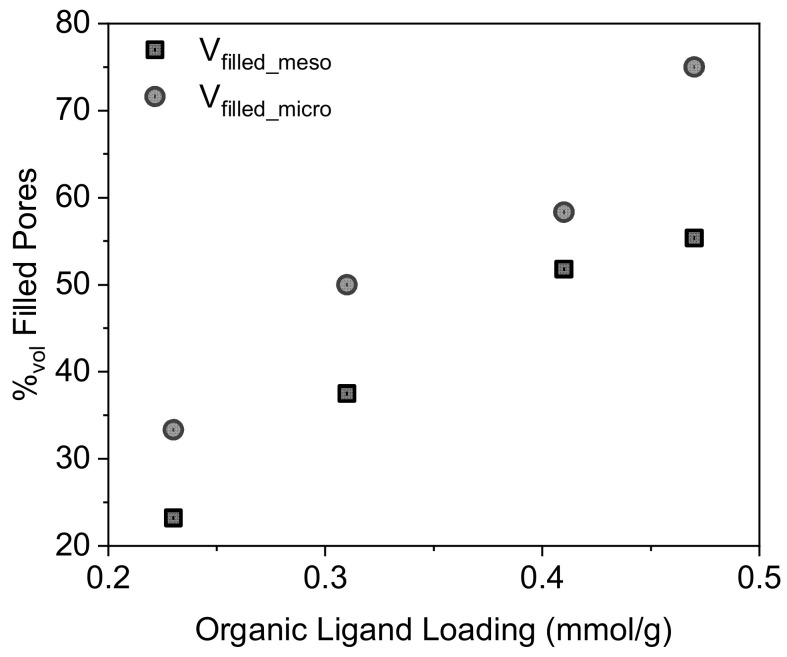
Volume percentage of filled micropores and mesopores after impregnation of the pre-functionalized support as a function of the concentration of organic ligand (DEHCEBP) in the materials.

**Figure 5 molecules-27-04342-f005:**
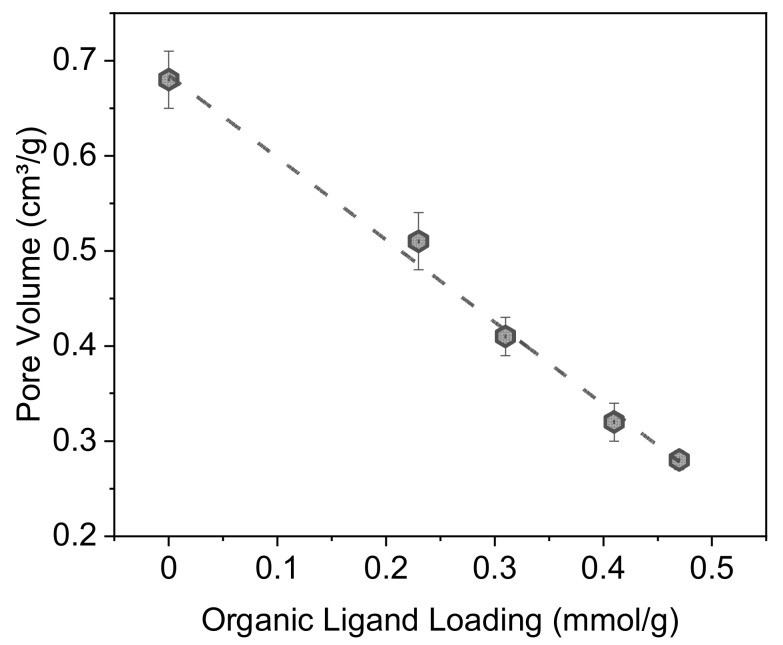
Residual pore volume in the pre-functionalized support and the impregnated materials as a function of the concentration of organic ligand (DEHCEBP) in the materials.

**Figure 6 molecules-27-04342-f006:**
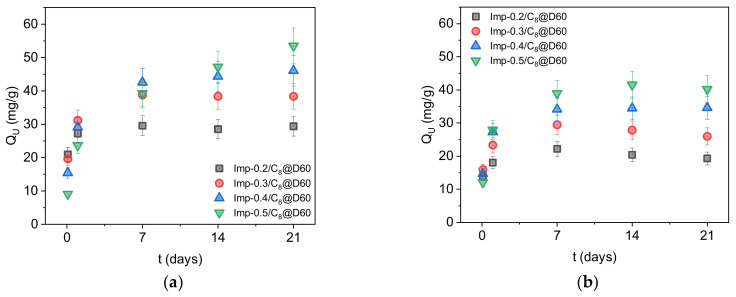
Uranium extraction capacity of the four functionalized materials with different ligand concentrations after 2 h and 1, 7, 14 and 21 days in sulfuric acid solutions ([SO_4_]^2−^/[U] = 900 mol/mol) (**a**) in the absence and (**b**) in the presence of competing cations (Fe and Mo).

**Figure 7 molecules-27-04342-f007:**
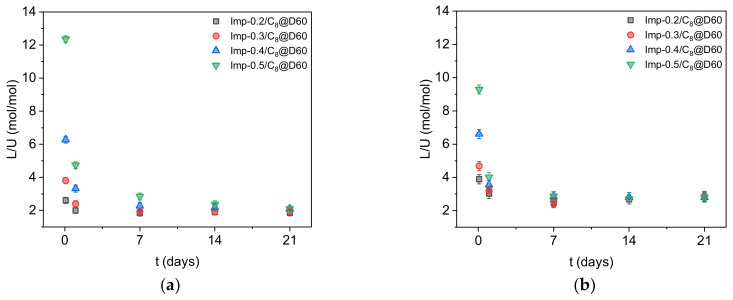
Ligand to uranium molar ratio of the four functionalized materials with different ligand concentrations after 2 h and 1, 7, 14 and 21 days in sulfuric acid solutions ([SO_4_]^2−^/[U] = 900 mol/mol) (**a**) in the absence and (**b**) in the presence of competing cations (Fe and Mo).

**Figure 8 molecules-27-04342-f008:**
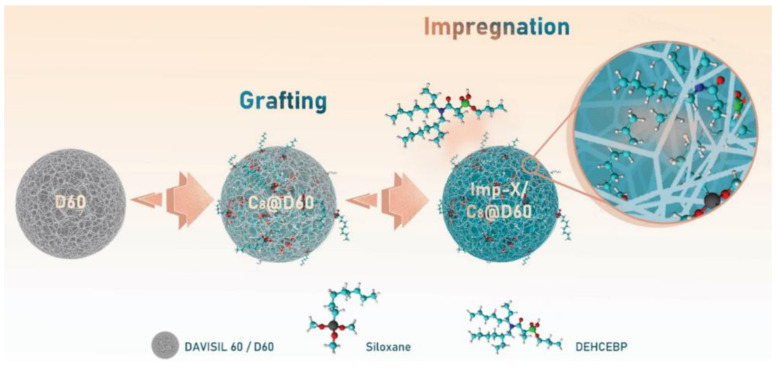
Schematic representation of the two-step process used to synthesize the amidophosphonate-impregnated materials.

**Table 1 molecules-27-04342-t001:** Properties of pure silica and the different materials synthesized in this study.

Material	*τ_L_* ^a^	%_L_ ^b^	*S_BET_ ^c^*	*V_pores_* ^d^	V_micropores_ ^e^	*V_L_* ^f^	*d_SLigand_* ^g^
C_8_@D60			447 (±22)	0.68 (±0.03)	0.12		
Imp-0.2/C_8_@D60	0.23 (±0.02)	9.9 (±0.9)	337 (±17)	0.51 (±0.03)	0.08	1.23	0.31
Imp-0.3/C_8_@D60	0.31 (±0.02)	13.7 (±0.9)	294 (±15)	0.41 (±0.02)	0.06	1.45	0.42
Imp-0.4/C_8_@D60	0.41 (±0.02)	17.6 (±0.9)	222 (±11)	0.32 (±0.02)	0.05	1.46	0.55
Imp-0.5/C_8_@D60	0.47 (±0.02)	20.4 (±0.9)	149 (±7)	0.28 (±0.01)	0.03	1.41	0.63

^a^ Concentration (mmol∙g^−1^) of DEHCEBP measured by TGA using Equation (1). ^b^ Mass percentage of DEHCEBP. ^c^ Specific surface area (m^2^∙g^−1^). ^d^ Total pore volume, measured at P/P_0_ ≈ 1 (cm^3^∙g^−1^). ^e^ Micropore volume. measured at P/P_0_ ≈ 0.05 (cm^3^∙g^−1^). ^f^ Volume of one DEHCEBP molecule (nm^3^) using Equation (2). ^g^ Surface density of the ligand (DEHCEBP∙nm^−2^) using Equation (3).

**Table 2 molecules-27-04342-t002:** Initial compositions of the uranium-containing simulated ore leaching solutions.

Elements Present	U (mg/L)	Fe (mg/L)	Mo (mg/L)	(SO_4_)^2−^ (g/L)	pH
U only	400	-	-	146	1
U + competing ions	400	5000	57	146	1

**Table 3 molecules-27-04342-t003:** Uranium extraction parameters of the functionalized silica materials.

Material	Elements Present in the Solution					
	U			U, Fe, Mo		
	Q_Umax_ ^a^	L/U ^b^	t ^c^	Q_Umax_ ^a^	L/U ^b^	t ^c^
Imp-0.2/C_8_@D60	30 (±3)	2.0 (± 0.2)	0–1	22 (±2)	2.5 (±0.3)	0–1
Imp-0.3/C_8_@D60	39 (±4)	1.9 (± 0.2)	1–7	29 (±2)	2.5 (±0.3)	1–7
Imp-0.4/C_8_@D60	46 (±5)	2.1 (± 0.2)	1–7	35 (±3)	2.8 (±0.3)	1–7
Imp-0.5/C_8_@D60	54 (±5)	2.1 (± 0.2)	14–21	42 (±4)	2.7 (±0.3)	1–7

^a^ Maximum uranium extraction capacity (mg∙g−1). ^b^ Ligand to uranium molar ratio (mol/mol). ^c^ Estimated equilibration time (days).

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
