# Peer review of "Effects of Impregnated Amidophosphonate Ligand Concentration on the Uranium Extraction Behavior of Mesoporous Silica"

_molecules, 2022, doi:10.3390/molecules27144342_

Round 1

Reviewer 1 Report

The present manuscript deals with the “Effects of impregnated amidophosphonate ligand concentration on the uranium extraction behavior of mesoporous silica”. In other to improve the manuscript quality I have a few questions, suggestions and recommendations.

1.       A better highlight of the research goals has to be inserted to the introduction part.

2.       How the pore size distribution influences the uranium extraction capacity? Is there any connection between the smaller pore size and the better extraction capacity of the Imp-0.5/C8@D60 sample compared to other samples, or it is related only to the ligand to uranium molar ratio? Please discuss.

3.       Unfortunately, I cannot find the supplementary materials. I would like to ask you if there are any interaction between the silica support and the ligand materials? Did you record NMR spectra regarding to those cases?

4.       Several “Error! Reference source not found” are present in the text. Please correct them.

5.       The figure 7 is present four times in between page 8 and 9.

6.       Please correct figure 77b present at lines 236 and 240

7.       At the references majority of the journals are presented by their full name, please correct the abbreviated ones.

Reviewer 2 Report

The paper by Aline Dressler and co-author describes a synthesis of a series of solid phase uranium extractants by post-synthesis impregnation of a mesoporous silica support with different loadings of an amidophosphonate ligand (DEHCEBP). The obtained Imp/C8@D60 composites was characterized through FTIR, 13C, 31P and 29Si CP-MAS NMR, TGA, and Nitrogen adsorption-desorption isotherms. However, the paper needs to be more carefully revised and both some detailed experiments and sufficient description of the relative results need to be supplemented. For these reasons, I cannot recommend publication of the present work in "Molecules". Specific Comments:

 (1) It is well known that the PH value is an important facet for the adsorption of metal ions. Therefore, I wonder if the PH value will affect the uranium extraction behavior of Imp/C8@D60 composites or not?

(2) TEM images for Imp/C8@D60 composites should be provided for further characterize their inner structure.

(3) The uranium extraction performance of the present work should be compared with some earlier reports.

(4) More experiment data should be provided for explaining the uranium extraction mechanisms.

(5) In page 4 line 133, authors mentioned that the TGA data can be found in supporting material – I. However, no any supporting materials were provided in paper.

(6) There are many misspelling and formatting mistakes for Figures and content of manuscript. Please check and revise.

Reviewer 3 Report

This article is very well written, research results are discussed in depth and the flow of discussion is arranged systematically. The novelty element in the article is sufficient and is in accordance with the state of the art research on materials. A few notes, research results need to be presented quantitatively both in the abstract and in the conclusion. It is also necessary to compare the research results achieved with the results of other researchers who also conducted similar research.

Round 2

Reviewer 2 Report

The authors have addressed the comments and the paper can be accepted.